# Proximate, Vitamins, Minerals and Anti-Nutritive Constituents of the Leaf and Stem of *Helichrysum odoratissimum* (L.) Sweet: A Folk Medicinal Plant in South Africa

Abolaji Olajumoke Afuape [1], Anthony Jide Afolayan [1] and Lisa Valencia Buwa-Komoreng [1,*]

Department of Botany, University of Fort Hare, Alice 5700, South Africa
* Correspondence: lbuwa@ufh.ac.za; Tel.: +27-406-02-2648

**Abstract:** *Helichrysum odoratissimum* (L.) Sweet (Asteraceae) is a perennial shrub used in South African folk medicine to combat human diseases such as diabetes, coughs, asthma, and diarrhoea, and for wound treatment. This study evaluated the leaf and stem of *H. odoratissimum* for their nutritive and anti-nutritive qualities using the standard methods of the Association of Official Analytical Chemists (AOAC), atomic absorption spectrometry (AAS) and colorimetry. Crude fat and protein were high in the leaf (7.61% and 7.82% DW), but low in the stem (2.25% and 3.4%), respectively. The neutral detergent fibre (NDF) content was significantly higher in the stem (68.5%) compared to the leaf (43.6%), while non-fibre carbohydrate (NFC) in the leaf (24.93%) was higher than in the stem (15.67%). Mineral analysis showed that both the leaf and stem of *H. odoratissimum* are good sources of potassium, calcium, sodium, magnesium, phosphorus and iron. Appreciable amounts of vitamins B2, C and E in the leaf and stem supports the folkloric utility of the plant in the traditional treatment of wounds, coughs and colds. A tolerable amount of phytate in both leaf (0.36%) and stem (1.06%) might be a factor for safer consumption of the plant as food and medicine. These findings suggest the use of the plant as a source for dietary supplementation and ethnomedicinal applications.

**Keywords:** folk medicine; *Helichrysum odoratissimum*; ICP-OES; minerals; phytate

## 1. Introduction

Phytotherapy continues to play a vital role in the treatment and management of human diseases in most parts of Africa, where there are limited resources and most people do not have access to conventional treatments [1–3]. This is very true of South Africa, where a large proportion of the people still live below the poverty datum line and with limited access to modern healthcare, thus relying on traditional medicine for managing chronic and noncommunicable diseases such as diabetes, obesity and their complications [3–5]. In recent times, the study of plant foods with nutraceutical, pharmaceutical, nutritional and functional potentials has been on the increase, and many plants with multifunctional properties are now gaining recognition and usage [4,6–10]. Due to a lack of knowledge about some of these multipurpose medicinal food plants which are effective resources to combat malnutrition and meet the required dietary intake, such plants are underutilized [7]. One such medicinal plant is *Helichrysum odoratissimum*, which has become a subject of scientific study due to its myriads of folkloric utility in traditional medicine.

*H. odoratissimum*, also known as the everlasting plant, is one of the important medicinal plants of Asia, Europe, Madagascar and general Africa. In South Africa, it is an important indigenous medicinal plant used in folk medicine across provinces such as Free State, KwaZulu-Natal, Limpopo, and Eastern and Western Cape [11,12]. The beneficial properties of this plant are attributable to the inherent chemical compositions such as flavonoids, phenolics, proanthocyanidins and tannins [11,12]. Traditionally, it is known for the treatment of wounds, coughs, colds, chest pain, diabetes and insomnia [13,14]. The leaf extract is used as a prophylactic for skin cancer [15], decoction and infusions of the leaf and stem are

taken orally in the traditional treatment of diabetes [13] and leaf decoctions are used for the treatment of pimples [16].

Pharmacological studies of *H. odoratissimum* crude leaf extracts reveal that the species possess antibacterial, antifungal, antiglycemic, anti-inflammatory, antimycobacterial, and antioxidant properties [17,18]. The anti-acne potential of the leaf and stem has also been reported [19]. Despite extensive pharmacological studies, the nutritional and anti-nutritional qualities of this plant are yet to be explored. Hence, this study evaluated the nutritive and anti-nutritive factors in the leaf and stem of *H. odoratissimum*, to further justify and authenticate its usage and medical application in folk medicine.

## 2. Materials and Methods

### 2.1. Plant Collection and Preparation

Fresh leaves and stem of *H. odoratissimum* were collected from sloppy temperate grassland vegetation (Velds) in Binfield Park Dam, latitude 32.6869° S and 26.9044° E, along Hogsback road, Raymond Mhlaba Municipality, Eastern Cape, South Africa, in late autumn (in May). The plant sample was authenticated by Professor Cupido, a taxonomist in the Botany Department at the University of Fort Hare. A voucher specimen (UFH 2021-10-01) was deposited in the Giffen Herbarium of the Department. The leaf and stem samples were washed with distilled water to remove dirt, dried gently with a laboratory paper towel, chopped into small pieces and then dried in an oven (LABOTEC, Cape Town, South Africa) at 50 °C for 120 h to a constant weight. An automated motor blender (Polymix® PX-MFC 90 D, Geneva, Switzerland) was used to pulverize the dried samples into a fine powder, which was stored in airtight glass bottles at 4 °C until further analysis. All reagents and chemicals used in this study were of analytical grade and were purchased from Sigma-Aldrich (St. Louis, MO, USA).

### 2.2. Proximate Content Analysis

Powdered *H. odoratissimum* leaf and stem were evaluated for moisture, ash, crude fat (CF), crude protein (CP) and crude fibre on a 100% dry matter basis using the Association of Official Analytical Chemists, AOAC [20] methods.

#### 2.2.1. Estimation of Moisture Content

Exactly 5 g each of the samples was placed in preweighed crucibles (Wa), weighed (Wb) and dried in an oven (LABOTEC, Cape Town, South Africa) at 105 °C for 24 h. This was allowed to cool in a desiccator and reweighed (Wc). The percentage moisture content was calculated as described in Equation (1):

$$\% \text{ Moisture content } = \left\{ \frac{Wb - Wc}{Wb - Wa} \right\} \times 100 \tag{1}$$

where Wa = weight of empty crucible, Wb = weight of empty crucible + sample, and Wc = weight of crucible + dried sample.

#### 2.2.2. Estimation of Ash Content

A clean porcelain crucible was dried in a muffle furnace at 105 °C for 1 h, cooled in a desiccator and weighed (Xa). One gram of each of the pulverized samples was placed in porcelain crucibles, weighed (Xb) and incinerated in the muffle furnace (Furnace E-Range, E300-P4, MET-U-ED, Johannesburg, South Africa), ashed first at 250 °C for 1 h, then at 550 °C for 6 h to assure complete ashing. The ashed samples in crucibles were removed, allowed to cool in a desiccator and then weighed (Xc). The percentage ash was estimated using Equation (2):

$$\% \text{ Ash content } = \left\{ \frac{Xc - Xa}{Xb - Xa} \right\} \times 100 \tag{2}$$

### 2.2.3. Estimation of Fat Content

Exactly 2 g of each of the pulverized dried samples was wrapped in filter paper and placed in a lipid-free thimble and added to the extraction tube. A preweighed clean beaker (M1) containing 100 mL of petroleum ether was fitted to the apparatus, and then the water and heater were turned on to start the extraction process. After five rounds of siphoning, ether was allowed to evaporate and the beaker was disconnected before the last siphoning. The extract was concentrated to dryness in a steam bath, oven-dried at 50–60 °C and then the beaker was reweighed (M2). The percentage of crude fat was estimated using Equation (3):

$$\% \text{ Crude Fat} = \left\{ \frac{M2 - M1}{\text{weight of plant sample}} \right\} \times 100 \qquad (3)$$

### 2.2.4. Estimation of Crude Protein Content

The crude protein of the leaf and stem of *H. odoratissimum* was estimated using the Kjeldahl method which involves digestion, distillation and titration (AOAC, [20]). The nitrogen content of the plant samples was measured first and multiplied by 6.25 to get the protein content. A factor of 6.25 was used because most protein contains approximately 16% nitrogen. Exactly 2 g of each of the pulverized leaf and stem samples of *H. odoratissimum* was placed in a 300 mL Kjeldahl flask and digested with 20 mL of concentrated sulphuric acid and 8 g of digestion mixture ($K_2SO_4$, $CuSO_4$ [8:1]) until a clear mixture was achieved. The digest was cooled, transferred into a volumetric flask (100 mL) and diluted with deionised water to make up to mark. 10 mL of the digest was put into the distillation tube and 10 mL of 0.5 N sodium hydroxide was added slowly into the tube. 2 to 3 drops of methyl red were added to 250 mL of 2% boric acid in a flask and the condenser containing the distillate was immersed inside this liquid to trap the ammonia gas. Thereafter, the collecting liquid was titrated against 0.01 M hydrochloric acid and a violet colour indicated the endpoint. A blank was also run following the same procedure as above. The percentage of nitrogen content was calculated using Equation (4).

$$\% \text{ Nitrogen (N)} = \left\{ \frac{(s - b) \times N \times 0.014 \times D}{w \times v} \right\} \times 100 \qquad (4)$$

where 's' and 'b' denotes the titration readings of the sample and blank, respectively; The constant 0.014 is the milli-equivalent weight of nitrogen, 'N' is the Normality of hydrochloric acid, 'D' is the dilution of the sample after digestion, 'w' is the weight of the sample and 'v' is the volume of the distilled digest.

The percentage of crude protein was expressed in Equation (5):

$$\% \text{ Crude Protein} = \% \text{ Nitrogen} \times 6.25 \qquad (5)$$

Neutral detergent fibre (NDF) and acid detergent fibre (ADF) were estimated according to the methods of Soest and Robertson [21]. Total non-fibre carbohydrate (NFC) was calculated as $100 - [(\text{NDF-ADF}) + \text{CP} + \text{Fat} + \text{Ash}]$. The energy value of the two samples was determined using the Atwater general factor system [22]. This is achieved by multiplying the crude fat content by 9%, crude protein by 4% and total carbohydrate by 4%. The energy content was then calculated as energy content = (% CF × 9) + (% CP × 4) + (% CHO × 4), where CHO is the total carbohydrate. All the analyses were performed in three replicates.

### 2.3. Vitamin Contents Evaluation

Vitamins $B_1$ (Thiamine), $B_2$ (Riboflavin) and $B_3$ (Niacin) were estimated according to the method of Okwu et al. [23]. Vitamin C (Ascorbic acid) and vitamin A (retinol) were determined according to the method outlined by Adegbaju et al. [24]. The vitamin E (alpha-tocopherol) content of the dried leaf and stem of *H. odoratissimum* was estimated

spectrophotometrically using the method described by Idris et al. [25]. All the analyses were performed in three replicates.

### 2.4. Mineral Element Analysis

Adopting the method of Okalebo, [26], the inductively coupled plasma optical emission spectrometer (ICP-OES; Varian 710–ES series, SMM Instruments, Cape Town, South Africa) was used to analyse the dried plant samples for iron content, while an atomic absorption spectrophotometer (Varian Spectra AA-220, Washington, DC, USA) was used to determine the calcium, copper, magnesium, manganese, phosphorus, potassium, sodium, and zinc contents. All the analyses were performed in three replicates.

### 2.5. Anti-Nutrient Content Estimation

The amount of oxalate in the leaf and stem samples was determined using the titration method outlined by Adegbaju et al. [24], while the phytate content of the leaf and stem samples was determined using the spectrophotometric method of Lalitha Sree and Vijayalakshmi, [27]. All the analyses were performed in three replicates.

### 2.6. Statistical Analysis

Data obtained from this study were processed with Microsoft Excel 2010 and presented as mean $n = 3 \pm$ standard deviation. The MINITAB version 17 statistical package was used to compare means using Fischer's least significance difference (LSD) at $p < 0.05$ significant difference.

## 3. Results

### 3.1. Proximal Content

The proximate content of the leaf and stem of *H. odoratissimum* is presented in Table 1. The moisture and ash contents of the leaf (6.01%; 10.07%) were higher than in the stem (4.86%; 5.27%). Also, crude fat in the leaf (7.61%) was three times that of the stem (2.25%), while the crude protein (7.82%) was found to be twice that of the stem (3.40%). The fibre content was estimated as acid detergent fibre (ADF) and neutral detergent fibre (NDF). The ADF and NDF values of the stem (47.27% and 68.54%) were significantly higher than that of the leaf (32.31% and 46.56%), respectively. The carbohydrate was estimated as non-fibre carbohydrate (NFC) by deducting the sum of crude protein, NDF-ADF, crude fat and ash from 100. NFC was higher in the leaf (34.13%) than in the stem (15.67%). The estimated oxidizable energy of the leaf (199.46 KJ/100 g) was significantly higher than that of the stem (96.54 KJ/100 g) at $p < 0.05$.

**Table 1.** Proximate composition of *H. odoratissimum* leaf and stem (%) of dry weight.

| Parameters | Leaf | Stem |
| --- | --- | --- |
| Moisture | 6.01 ± 0.22 [a] | 4.86 ± 0.02 [b] |
| Ash | 10.07 ± 0.26 [a] | 5.27 ± 0.22 [b] |
| Crude fat | 7.61 ± 0.39 [a] | 2.25 ± 0.19 [b] |
| Crude fibre | 43.56 ± 2.77 [b] | 68.54 ± 0.91 [a] |
| Crude Protein | 7.82 ± 0.38 [a] | 3.40 ± 0.17 [b] |
| ADF | 32.31 ± 0.09 [b] | 47.27 ± 1.59 [a] |
| NDF | 46.56 ± 0.01 [b] | 68.54 ± 0.91 [a] |
| Carbohydrate (NFC) | 34.13 ± 0.02 [a] | 15.67± 0.35 [b] |
| Energy value (KJ/100 g) | 199.46 ± 9.72 [a] | 96.54 ± 3.74 [b] |

Values are expressed as three replicates of mean ± S.D; [a] and [b] are superscripts indicating significant difference ($p < 0.05$); values within the same row with different superscripts are significantly different. ADF, acid detergent fibre; NDF, neutral detergent fibre; NFC, non-fibre carbohydrate.

### 3.2. Vitamins and Antinutrients

The vitamins and anti-nutrient contents are shown in Table 2. The stem exhibited significantly high values for vitamins A, B1, B2 and C, while the leaf was significantly high in vitamin E. The vitamin B3 content of the leaf and stem of the studied plant were not significantly different at $p < 0.05$. Among the vitamin B complex family, vitamin B1 was highest for both the leaf and stem (24.59 mg/100 g and 26.39 mg/100 g), followed by vitamin B2 (0.62 mg/100 g and 2.34 mg/100 g) and B3 (0.18 mg/100 g and 0.095 mg/100 g), respectively. The vitamin C content was (15.65 mg/100 g) and (17.55 mg/100 g) in the leaf and stem, respectively. Significant amounts of vitamin A (373.5 µg/100 g and 489.5 µg/100 g) and vitamin E (2.25 µg/100 g and 0.92 µg/100 g) were also present in both leaf and stem, respectively. Oxalate was present only in the stem (6.05%) but absent in the leaf, while phytate was found in a trace amount for both the leaf (0.36%) and stem (1.06%).

**Table 2.** Vitamin and anti-nutrient content in the leaf and stem of *H. odoratissimum*.

| Vitamin (mg/100 g) DW | Leaf | Stem |
|---|---|---|
| Retinol (A) (µg) | 373.5 ± 1.25 [b] | 489.5 ± 9.67 [a] |
| Thiamine (B$_1$) | 24.59 ± 0.01 [b] | 26.39 ± 2.14 [a] |
| Riboflavin (B$_2$) | 0.62 ± 0.001 [b] | 2.34 ± 0.001 [a] |
| Niacin (B$_3$) | 0.18 ± 0.0004 [a] | 0.095 ± 0.0002 [a] |
| Ascorbic acid (C) | 15.65 ± 1.11 [b] | 17.55 ± 0.35 [a] |
| Alpha-tocopherol (µg) | 2.25 ± 0.001 [a] | 0.92 ± 0.001 [b] |
| **Anti-nutrient (%)** | | |
| Oxalate | 0 ± 0 | 6.05 ± 0.08 |
| Phytate | 0.36 ± 0.04 | 1.06 ± 0.23 |

Values are expressed as triplicates of mean ± S.D; Superscripts [a] and [b] indicate a significant difference ($p < 0.05$); values within the same row with the same superscript are not significantly different. DW: dry weight.

### 3.3. Mineral Element Composition

A comparative mineral profile of the leaf and stem of *H odoratissimum* is presented in Table 3. The leaf sample displayed significantly high values of all the macro- and microelements analysed in this study. The values of phosphorus, manganese, copper and zinc in both leaf and stem were significantly the same. Both the leaf and stem were high in potassium content (11.13 mg/100 g and 7.87 mg/100 g), respectively, but low in phosphorus content (0.03 mg/100 g and 0.73 mg/100 g), respectively. On the other hand, both samples were high in iron content (107.3 mg/100 g and 70.7 mg/100 g), but low in copper content (2.33 mg/100 g and 1.37 mg/100 g), respectively. However, the K/ Ca+ Mg ratio was higher in the stem (0.45 mg/100 g) than in the leaf (0.37 mg/100 g).

**Table 3.** Mineral composition of *H. odoratissimum* leaf and stem (mg/100 g) DW.

| Elements | Leaf | Stem |
|---|---|---|
| Calcium (Ca) | 9.07 ± 0.05 [a] | 5.67 ± 0.45 [b] |
| Magnesium (Mg) | 3.73 ± 0.19 [a] | 2.03 ± 0.09 [b] |
| Potassium (K) | 11.13 ± 0.62 [a] | 7.87 ± 0.29 [b] |
| Sodium (Na) | 7.53 ± 0.62 [a] | 2.83 ± 0.12 [b] |
| Phosphorus (P) | 0.03 ± 0.05 [a] | 0.73 ± 0.05 [a] |
| K/Ca+ Mg | 0.37 ± 0.01 [b] | 0.45 ± 0.02 [a] |
| Zinc (Zn) | 5.67 ± 0.05 [a] | 4.13 ± 0.12 [a] |
| Manganese (Mn) | 9.03 ± 1.56 [a] | 8.13 ± 0.58 [a] |
| Copper (Cu) | 2.33 ± 0.52 [a] | 1.37 ± 0.21 [a] |
| Iron (Fe) | 107.3 ± 31.5 [a] | 70.7 ± 15.8 [b] |

Values are expressed as triplicates of mean ± S.D; Superscripts [a] and [b] indicate a significant difference ($p < 0.05$); values within the same row with the same superscript are not significantly different. DW: dry weight.

## 4. Discussion

Proximate analysis of medicinal plants is crucial in determining the type and amount of chemicals present in them to enhance their usage and authenticity. *H. odoratissimum* is an indigenous medicinal plant used in folkloric medicine for various ailments. The present study reveals that this plant contains appreciable amounts of nutrients, minerals, vitamins and safe levels of antinutrients.

Reports suggested that the moisture contents of plants vary based on the species and plant parts under study [28]. The difference in the moisture contents of the leaf and stem of *H. odoratissimum* agrees with the report conducted on the leaf and stem of *Costus afer* [29]. Low moisture content below 10% found in both plant samples is a very good attribute for medicinal plants as this factor will serve as a preservative measure against microbial growth during storage and prevent spoilage [30,31].

The low crude fat content of *H. odoratissimum* stem compared favourably with reports for most medicinal plants such as *Bryophyllum pinnatum* (0.56%), *Alocasia indica* (1.44%), and *Urena lobata* (1.7%) [15,32]. This may suggest the importance of this plant as a resource for an antiobesity dietary supplement [6,33].

Protein is crucial for body fluid maintenance, sustaining a strong immune response, and the synthesis of enzymes and hormones [33]. The crude protein content in the leaf, though higher than that of the stem, was still relatively low when compared with the normal protein value of greater than 20% in the diet as recommended by the National Research Committee on Dietary allowance [34]. However, the moderate available protein content of the plant could be a source of dietary protein and fats for feed supplements [24,30].

High ash content is an indication of a high level of mineral concentration in the plant [30,31]. This agrees with the higher mineral concentration in the leaf of this plant sample (Table 3). The ash content of 10% obtained from the leaf compared favourably with the reports for other medicinal plants such as *Cassia sophera* (10.96%), *Hypoxis hemerocallidea* (14.85%), and *Tridax procumbens* (4.3%) [6,35]. The ash content also implies that the plant could act as a preservative (inhibiting the growth of microorganisms), a digestion aid, and could be a good source of inorganic minerals [29,33].

Neutral detergent fibre (NDF) and Acid detergent fibre (ADF) are important measurements used in forage feed for animals [36]. Neutral detergent fibre (NDF) is used to estimate the content of fibre in cellulose, hemicellulose, lignin, cutin and insoluble minerals in the cell wall, while acid detergent fibre (ADF) is used to estimate the content of fibre in cellulose, lignin, cutin and insoluble minerals in the cell wall [37,38]. The two fibre calculations are used to determine the amount (quality) and energy that will be contained in animal feed [36]. The difference between the two fibre estimators is a fraction of the hemicellulose that is included in the estimation of neutral detergent fibre [36,38]. Fibres with low lignin, cellulose and hemicellulose will occupy less space in the stomach and provide a huge amount of energy to animals [38]. The whole fibre fraction of a feed is determined as neutral detergent fibre (NDF). It is regarded as the structural component of a plant which predicts the voluntary intake of the bulk or fills [39]. The crude fibre contents of the stem are higher than that of the leaf and this supports the findings of Anyasor et al. [29] and Datta et al. [40]. The low NDF value of the leaf connotes that it will be easily digested and this suggests that the plant could serve as a resource for dietary fibre. The least digestible plant components are inherent in acid detergent fibre (ADF), which is inversely related to digestibility. A low ADF value is attributable to a higher energy value; this suggests that the leaf is likely to increase the energy value of any diet it is added to compared to the stem [6]. Previous studies reported that dietary fibre could regulate intestinal transits and reduce the risk of various metabolic disorders, such as diabetes and its health risk factors including hypertension, atherosclerosis and obesity [28–30,40].

Plants synthesize, store and break down carbohydrates into simpler molecules to generate the required energy for growth and metabolism in both plants and animals [41]. The leaf and stem of *H. odoratissimum* contain an appreciable amount of digestible carbohydrates that could provide both animals and the human body the necessary energy required

to promote cellular metabolism [41]. The oxidizable energy levels in the present study ranged from 96.54 KJ/100 g in the stem to 199.46 KJ/100 g in the leaf. The energy values of selected medicinal plants in India fall within this range [41]. A similar calorific value has been obtained in scientific studies, with references to *Heteromorpha arborescens* (Spreng) [7]. A plant with a high oxidizable energy value can be considered a good supplement to the diet [40–42], hence, *H. odoratissimum* could be a good source of dietary supplements.

Minerals are crucial components of hormones, vitamins, enzyme activation systems, and/or cofactors in cellular metabolism [30]. Appreciable amounts of potassium, sodium, calcium and magnesium were observed in both the leaf and stem of *H. odoratissimum*, though higher in the leaf than in the stem. Potassium, sodium and magnesium are needed in the body for the control, treatment and management of various metabolic and heart disorders [43,44].

The calcium content in the plant sample confers the capacity to strengthen the bones and teeth, maintain extracellular fluids, and transmit nerve impulses, blood clotting and muscle contraction [33,41]. The low phosphorus content reported in the leaf and stem indicates that *H. odoratissimum* is a poor source of phosphorus. This result agrees with reports of the low phosphorus content in other medicinal plants such as *Acanthus montanus*, *Hypoxis hemerocallidea* and *Dendrobium officinale* [6,45]. Although phosphorus plays a crucial role in metabolism and growth, its low concentration in plants has been found useful for the improvement in the bioactive ingredient and free radical scavenging capacity in medicinal plants [46]. The importance of lower phosphorus in potential dietary strategies, as one of the therapeutic targets for the management of chronic kidney diseases, mineral and bone disorders, has been reported [47]; thus, *H. odoratissimum* could be functional in the management of chronic kidney diseases.

Iron is a crucial element required for the biosynthesis of haemoglobin in erythrocytes and energy, as well as in the physiology of oxygen transportation [24]. Appreciable amounts of iron in the leaf and stem of *H. odoratissimum* support the traditional use of the plant to treat anaemia [30,40]. Zinc is crucial in protein formation, the differentiation and replication of cells, immunity, sexual functions and as a cofactor in nucleic acid and carbohydrate metabolism [24,48]. Copper is involved in the metabolism of iron and the synthesis and maintenance of erythrocytes, while manganese functions in enzyme activation, osteogenesis, erythrocyte rejuvenation and glucose metabolism [30]. Appreciable amounts of trace elements recorded in this plant suggest that the plant could be a good resource for food supplementation and new drug discovery. The K/Ca+Mg ratio of the samples analysed was low compared with the standard requirement ratio of 2 [49]. Magnesium and calcium antagonize each other in reabsorption, inflammation and many other physiological processes. However, the low K/Ca+Mg ratio reported in this study showed that reabsorption will not be altered because dietary magnesium, calcium, zinc and phosphorus will be readily available in normal proportions [6,49]. Antinutrients in plants provide self-defence and optimally decrease the release and absorption of nutrients such as protein, vitamins and minerals [50]. Oxalate is minimally present in the stem but absent in the leaf. The regulation of calcium, resistance to herbivores, heavy metal tolerance and facilitation of plant growth, are some of the biological functions of oxalate in plants [51]. Phytate has been reported to have effects on the intake and adequate usage of important nutrients in the human body [50]. The lower phytate content in the leaf and stem enumerate the increased absorption and utilization of minerals in humans [27]. The low oxalate content and low phytate levels recorded in this plant conform to the findings of Abifarin et al. [7] and Ebu et al. [50]. The values were below the lethal dose to inflict any injury on humans [25]. These add value to the medicinal plant analyses making them more acceptable and dependable.

Vitamins are organic compounds essential in minute quantities for healthy human growth. The present study showed that the leaves and stem of *H. odoratissimum* contain appreciable amounts of alpha-tocopherol, ascorbic acid, niacin, retinol and riboflavin. The presence of retinol in *H. odoratissimum* could confer on it the functionality of maintaining a

healthy body and normal vision [51]. Alpha-tocopherol, a lipophilic-free radical scavenger found in this plant, helps to protect sebaceous fatty acids from oxidative stress-related free radical damage [52]. Oxidation has been implicated in health conditions or diseases such as ageing, arthritis, obesity, diabetes and cancer [30,53]. Thus, the leaf and stem of *H. odoratissimum* might function in prophylactic measures against the aforementioned diseases. The leaf and stem of *H. odoratissimum* also contain appreciable amounts of hydrophilic vitamins. These vitamins function primarily as enzyme cofactors in various biological reactions [27,40]. Ascorbic acid (vitamin C) functions as an antioxidant and enzyme cofactor and helps to reduce oxidative stress, regulates immunological response, inhibits infections, aids wound healing, and relieves coughs and colds [37,53]. Ascorbic acid is readily available in the leaf and stem of *H. odoratissimum*. This justifies the folkloric use of this plant in the traditional treatment of wounds, coughs and colds [14,54]. Thiamine ($B_1$) is required for carbohydrate metabolism, the synthesis of energy and the stimulation of appetite [27,40]. Riboflavin ($B_2$) is a water-soluble antioxidant that functions in energy production and plays a crucial role in oxidation–reduction reactions [27,54]. Niacin ($B_3$) aids digestion, lowers the risks of cardiovascular diseases and plays a crucial role in cellular metabolism [37,53]. Deficiencies of these micronutrients have been implicated in metabolic disorders associated with malnutrition [27,51]. This plant, therefore, could be a better alternative as a resource for vitamins to ameliorate malnutrition.

## 5. Conclusions

The results obtained from this study showed that nutrients and minerals are more abundant in the leaf than in the stem of *H. odoratissimum*, except for crude fibre and vitamin contents. The proximate composition revealed that the leaf and stem of *H. odo ratissimum* are rich in crude fibre and carbohydrates, and this gives a corresponding higher digestibility and energy content to this plant. Mineral composition analysis showed that the leaves and stems are good sources of potassium, calcium, sodium, magnesium, phosphorus and iron. Data obtained from the vitamin content analysis revealed the presence of an appreciable amount of hydrophilic and lipophilic antioxidant vitamins in the leaf and stem of *H. odoratissimum*, which shows that the plant could be a good antioxidant agent. A moderate amount of antinutrients were also present in the leaf and stem of this plant. This suggests that the leaf and stem of this plant could also be integrated as a functional food into the diet. These findings have provided a novel report on the nutritional evidence for the extensive ethnomedicinal utility of this plant in the treatment of numerous human ailments and various oxidative stress-related diseases including hypertension, obesity, diabetes, and cardiovascular diseases. Further study to assess the biological function of the leaf and stem is currently in progress.

**Author Contributions:** Conceptualization, L.V.B.-K. and A.O.A.; methodology, L.V.B.-K. and A.O.A.; validation, L.V.B.-K., A.J.A. and A.O.A.; formal analysis, L.V.B.-K., A.J.A. and A.O.A.; investigation, A.O.A.; resources, L.V.B.-K., A.J.A. and A.O.A.; data curation, L.V.B.-K. and A.O.A.; writing—original draft preparation, A.O.A.; writing—review and editing, L.V.B.-K., A.J.A. and A.O.A.; visualization L.V.B.-K., A.J.A. and A.O.A.; supervision, L.V.B.-K., A.J.A. and A.O.A.; project administration, A.O.A. All authors have read and agreed to the published version of the manuscript.

**Funding:** L.V.B.-K. is funded by the Govan Mbeki Research and Development Centre, the University of Fort Hare, and the National Research Foundation of South Africa (Grant number 128897).

**Institutional Review Board Statement:** Not applicable.

**Informed Consent Statement:** Not applicable.

**Data Availability Statement:** The data used to support the findings of this study are available upon request to the authors.

**Conflicts of Interest:** The authors declare no conflict of interest with respect to this study, the authorship, and the publication of this research article.

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
