# Peer review of "Proximate, Vitamins, Minerals and Anti-Nutritive Constituents of the Leaf and Stem of Helichrysum odoratissimum (L.) Sweet: A Folk Medicinal Plant in South Africa"

_2037-0164, doi:10.3390/ijpb13040037_

Round 1
Reviewer 1 Report
The article makes simple determinations on some of the most important nutritional constituents. But there are very important aspects that need to be corrected:
Line 12: it is not necessary to put in the abstract how the methodology has been carried out, this information is for the section on materials and methods.
Line 14. put a space in "and7.82%".
In the abstract: the results are all expressed in %, but it is necessary to specify whether they refer to fresh or dry matter. To use acronyms such as NDF or NFC, they must be described the first time they are mentioned.
It would also be important in the introduction to mention if there are studies on the content of minority components with functional activity such as phenols that may be responsible for the properties attributed to this plant.
Line 52. The authors state the objective of the work but must specify which factors have been studied and why they have been used.
Line 62. The starting material, leaf and stem were washed with water, dried with paper and finally dried in an oven at 50 ºC for two hours. No matter how carefully the material is cleaned, if it is left for 120 hours at 50 ºC, many of the components that provide biological activity will degrade. This point should be well argued by the authors.
Line 105. Authors should not repeat the concentration values already shown in the table. The same for the rest of the results.
The authors should explain in the discussion what NDF and ADF mean, in terms of the amount of more or less available sugars, i.e., whether or not they have hemicellulose or only cellulose, as well as whether or not they have lignin, which hinders their metabolism and therefore their nutritional utilization. This point is very important in order to understand the points that have been discussed.
In order to establish the true nutritional importance, it is necessary to carry out some kind of digestion, since if the hemicellulose content is high but it is not very accessible due to the lignocellulosic framework in which it is found, it will not be of great nutritional interest. That is why limiting oneself to analyzing the composition can only give an idea of what is potentially there, but one can never affirm that it is nutritionally adequate because no digestion study has been carried out.
Line 186 talks about high and low NDF values, when the most digestible carbohydrates are more related to the difference between NDF and ADF, i.e. hemicellulose content, this difference has not been evaluated or discussed.
In general, when minerals and vitamins are presented in the discussion, a lot of activity data are presented that should be in the introduction, not in this section where only the values obtained should be compared and their importance explained. Neither can we talk in the discussion of components that have not been characterized in the work, such as tocopherols, among others. Therefore, the discussion should be simplified and focus on the parameters determined, comparing them with other works.
In the conclusion, line 299: fiber is carbohydrates, so it cannot be put as something different.
It can only be concluded in this work the content in the analyzed components, and it cannot be concluded if they are digestible or not, in any case it could be suggested that they can be, but that digestibility tests would be necessary.
Reviewer 2 Report
Dear Authors,
The article review: „ Proximate, vitamins, minerals and anti-nutritive constituents of the leaf and stem of Helichrysum odoratissimum (L.) Sweet: A folk medicinal plant in South Africa”. The presented research results are very interesting and worth publishing after correction of the manuscript.
General comments:
· Statistical analysis needs to be repeated. One-way analysis of variance (ANOVA) is a statistical tool designed to test differences between at least three groups. In the presented study 2 groups were compared. Therefore an adequate test to compare the 2 groups should be used.
Abstract:
Line 14: … and7.82%) …
Rev. space
Line 4: …Helichrysum odoratissimum (L.) Sweet… and Line 9: …Helichrysum odoratissimum Linn. …
Rev: the same name (spelling) should be given in all manuscript
Keywords:
Line 23: … antinutrients …; Line 121: … anti-nutrients …
Rev: standardize spelling
Rev: maybe add: vitamins, minerals?
Materials and methods:
Line 56: … 2.1 Plant collection and preparation …
Rev: better characteristics of the location should be given – plant community, co-occurring plant species, natural or transformed habitat?
Rev: more detailed characterization of the reagents should be given
Results:
Line 114: … (199.46 ± 9.73 KJ/100g) …
Rev: in Table 1 is other SD
Line 114-115: … (199.46 ± 9.73 KJ/100g) was higher than that of the 114 stem (96.54 ± 3.74 KJ/100g) …
Rev: SD is unnecessary
Line 115: … P Ë‚ 0.05 …
Rev: ?; in Line 118 is p Ë‚ 0.05
Line 119: … NDF,neutral …
Rev: space
Line 126-129: … (24.59 ± 0.01 and 26.39 ± 2.14), followed by vitamin B2 (0.62 126 ± 0.001 and 2.34 ± 0.001) and B3 (0.18 ± 0.0004 and 0.095 ± 0.0002), respectively. The vita- 127 min C content was (15.65 ± 1.11) and (17.55 ± 0.35) in leaf and stem, respectively. Signifi- 128 cant amounts of vitamin A (373.5 ± 1.25 and 489.5 ± 9.67) …
Rev: SD is unnecessary
Line 142-147: … (11.13 ± 0.62 mg/100 g and 7.87 ± 0.29 mg/100 g) but low in phos- 142 phorus content (0.03 ± 0.05 mg/ 100 g and 0.73 ± 0.05 mg/100 g), respectively. On the other 143 hand, both samples were high in iron content (107.3 ± 31.47 mg/100 g and 70.7 ± 15.81 144 mg/100 g) but low in copper content (2.33 ± 0.52 mg/100g and 1.37 ± 0.21 mg/100g), re- 145 spectively. However, the K/Ca/Mg ratio was higher in the stem (0.45 ± 0.02 mg/100g) than 146 in the leaf (0.37 ± 0.01 mg/100 g). …
Rev: SD is unnecessary
Line 146: … K/Ca/Mg …; Table 3: … K: Ca: Mg …
Rev: should be unified
Line 200: … Heteromorpha arborescens (Spreng) …
Rev: in References is Heteromorpha arborescens (Spreng.)
Line 252; … The K/ Ca / Mg …; Line 255; K/Ca/Mg
Rev: requires correction
Line 267: … Abifarin et al. [5] and …
Rev: requires correction
References:
Line 328: … Patel, D. K., Prasad, S. K., …
Rev: remove space – applies to the entire section
Line 328: … 2012.2(4), …
Rev: requires correction – applies to the entire section
Line 329-332: …
Moradi, B., Moradi, B., Abbaszadeh, S., Shahsavari, S., Alizadeh, M., Beyranvand, F. The most useful medicinal herbs to treat 329 diabetes. Biomedical Research and Therapy, 2018. 5(8), 2538–2551. https://doi.org/10.15419/bmrat.v5i8.463 330 3.
Nyakudya, T. T., Tshabalala, T., Dangarembizi, R., Erlwanger, K. H., Ndhlala, A. R. The Potential Therapeutic Value of Medicinal Plants in the Management of Metabolic Disorders. Molecules 2020, Vol. 25, Page 2669, 25(11), 2669. …
Rev: why different format? – requires correction – applies to the entire section
Line 336: … Heteromorpha arborescens; Line 349: … Helichrysum; Line 369: … Kedrostis Africana; Line 373: … Celosia argentea …; Line 381, 383, 386, 388, 395, 397, 416 ……….
Rev: requires correction – all latin names – italics – applies to the entire section
Line 337: … 2021. Food Science & Nutrition, …
Rev: requires correction
Line 349: … In Industrial Crops and Products 2019. 138, p. 111471 …
Rev: requires correction
Line 376-377: … Rumex crispus l. Plants, 2019. 8(3). …
Rev: requires correction
Line 398: … G Dike. ~ 629 ~ Jour …
Rev: requires correction
Line 426: … In Planta Medica 2019. 85 (4), pp. 312–334. …
Rev: requires correction
The section is prepared carelessly.
See: Instructions for Authors (https://www.mdpi.com/journal/ijpb/instructions)
Reviewer 3 Report
The article sent me for review is prepared in accordance with all the rules of writing scientific papers intended for publication. And this is where the praise actually ends. The work deals with a very well-known topic and uses commonly known and applied methods for routine research. In a well-known plant, it presents the content of basic substances for it and for most other plants. The conclusions drawn are so general that they almost cannot be wrong. The only element of originality of this research may be the place of collection of the tested plant, as we know that the same plant collected in different places and at different times may have a different composition.
Generally, the work is rather weak in terms of its originality and innovation, as well as methods and researched plant materials, but it is not a weakness that absolutely disqualifies it from applying for publication just in this Journal.
In order for this article to be published, the following errors should be corrected and inaccuracies should be clarified. I present a list of them here in order of their appearance in the text:
1. The analytical methods used in the work are an absolute routine, as if they were transferred out from the production line of some plant producing "traditional folk medicines". Maybe this description can somehow be made more similar to a research work.
2. In the title of Table 1 it was written that the results were given in %, but it is not known what these percentages are, is it in relation to the dry weight of the raw material? This needs to be clarified.
3. In the descriptive part of point 3.2. no units were given for the entire set of numerical values.
4. Basically, only the oxalate and phytate content can be considered as highly exhibited here a set of anti-nutrients substances. Do they really play such a role? It is quite a relative term, especially in relation to phytates.
5. The determined amounts of vitamins are rather small. Can be concluded, on this basis, that these amounts determine the beneficial nutritional characteristics, and even more so about the potential characteristics of the use of this plant "in drug discovery".
6. In point 3.3. the ratio K/Ca/Mg is stated to be expressed as mg/100g, but in Table 3 the same is stated in %. And whether the ratio of two quantities expressed in the same units at all can be determined in any units? For me, such a ratio is a numerical quantity without units (unitless). Besides, how one number can determined the ratio of the three components. This absolutely requires an explanation, how it is expressed in your article.
7. The obtained results, despite their banality, do not require such extensive discussion. It is completely unnecessary in this section to present the mode of action of individual vitamins.
8. The conclusions contain information about the possibility of using the studied plant, "in disease-treatment intervention and new drug discovery". This has not been demonstrated in this work. This conclusion is unfounded.
9. In the References part, the manner of writing the doi number should be standardized, if it is given as a reference to https.
After carrying out the above changes, the work can be accepted for publication, bearing in mind, however, that it will not be of any outstanding quality.
Reviewer 4 Report
The work, although interesting, is incomplete from an experimental point of view. In detail, H. odoratissimum appears to contain various bioactive compounds in the various parts of the plants, including monoterpenes. So it would be interesting to complete the work with the analysis of these compounds and related chemical characterization, also with regard to the fruits.
Other notes:
2.1 Plant collection and preparation: when were the plants collected? Have samples from different vintages been analyzed?
Lines 26-32: the authors report "Phytotherapy continues to play a vital role in the treatment and management of human diseases in most parts of Africa, where there are limited resources and most people do not have access to conventional treatments".
Please, it would be appropriate to include references from experts and high-impact factor journals to support the treatment of diseases in Africa with phytotherapy, which plays a vital role.
Lines 32-34: you have only added one reference. Please add other references from experts in the fields relating to the use of plant foods with nutraceutical, pharmaceutical, nutritional and functional potentials.
Line 64: the samples are dried at 50 °C for 120 hours. Are you sure that these parameters and in particular the high temperatures do not degrade the bioactive compounds present, most of which are thermolabile?
In Table 1: Why are the standard deviation values ​​for ADF, NDF, and leaf carbohydrates so high? Values ​​are expressed as a percentage of what?
In Table 2: are the values ​​expressed as mg / 100 g of dry weight or fresh weight? Are the retinol values ​​that high 373.5 mg for the leaves?
Round 2
Reviewer 1 Report
The manuscript has been substantially improved, although there are still two important questions that the authors need to clarify. Although the authors state that in this paper they have considered that carbohydrates are not fibre, this is a major error since by definition carbohydrates are sugars whether they have a single sugar molecule or thousands. Therefore, fibre as a highly polymerised polysaccharide is also a carbohydrate.
Reviewer 2 Report
The section References is prepared carelessly.
See: Instructions for Authors (https://www.mdpi.com/journal/ijpb/instructions)

Reviewer 3 Report
The correction of the revieved and evaluated article is not done in a perfect way, but it is done in a good enough way to be accepted for publication. The article still requires significant editorial efforts and the correction of defects and shortcomings called a typographical errors.
Reviewer 4 Report
The authors partially replied to the questions, however they did not modify or insert them in the new version of the manuscript.
